# Utility of enhanced recovery after surgery protocols in reducing postoperative opioid use across different surgical specialties—An analysis of Iowa's billion pill pledge program

Wali U. Pirzada[1,2], Simran Shamith[2], Roland N. Leyson[2], Sara A. Khan[2], Sina Ramtin[1,3]*, Asif M. Ilyas[1,2,4]

1 Rothman Opioid Foundation, Philadelphia, Pennsylvania, United States of America, 2 Drexel University College of Medicine, Philadelphia, Pennsylvania, United States of America, 3 University of Texas, Austin, Texas, United States of America, 4 Rothman Orthopaedic Institute, Philadelphia, Pennsylvania, United States of America

* Sina.ramtin@austin.utexas.edu

## Abstract

### Introduction

The opioid epidemic in the United States poses a major public health challenge, particularly in the context of surgery and perioperative pain management. This study examines the effectiveness of the "Billion Pill Pledge" Enhanced Recovery After Surgery (ERAS) protocols implemented across nine Iowa hospitals, in reducing postoperative opioid prescriptions.

### Methods

A retrospective chart review was conducted on patients treated by 24 different surgeons at 9 Iowa hospitals from November 2022 to November 2023. Patients were divided into orthopaedic surgery (n = 120) and general surgery (n = 60) groups. Opioid quantities prescribed pre- and post-ERAS implementation were measured and converted to morphine milligram equivalents (MMEs). Statistical analyses included the Wilcoxon-Signed Rank test, Mann-Whitney U test, and Chi-Squared tests.

### Results

The mean pre-ERAS prescription was 341 MMEs (range: 25–7200 MMEs), which decreased to 151 MMEs (range: 25–2400 MMEs) post-ERAS implementation (p < .001), following all surgeries. Orthopaedic Surgery patients saw a mean 45% reduction in prescription size from 462 MMEs (range: 50–7200 MMEs) to 197 MMEs (range: 25–2400 MMEs) (p < .001), while General Surgery patients experienced a mean 38% reduction from 100 MMEs (range: 25–150 MMEs) to 60 MMEs (range: 25–150 MMEs) (p < .001). Mean percent reduction in prescription size was greater in

**Data availability statement:** The statistics in this study were calculated using data provided by a third party: Goldfinch Health Incorporated. The authors of this study do not have permission to share these third party data publicly. Data for this study are available upon request from Meredith Allan, Vice President of Program Success, Goldfinch Health Incorporated, via email (meredith.allan@goldfinchhealth.com) for researchers who meet the criteria for access to confidential data. Others may replicate the study findings entirely by directly obtaining data from Goldfinch Health Incorporated and following the methodology outlined in the paper.

**Funding:** The author(s) received no specific funding for this work.

**Competing interests:** The authors have declared that no competing interests exist.

the Orthopaedic Surgery cohort (45% versus 38%) (p = .002). No significant difference was observed in the mean percentage of prescribed MMEs leftover between the two cohorts (Orthopaedic 47% vs. General 59%), (p = .07). Orthopaedic Surgery patients had higher mean MMEs consumed (126 MMEs) than General Surgery patients (26 MMEs) (p < 0.001).

## Conclusion

Postoperative opioid prescriptions were reduced in both patient surgical cohorts, with a more pronounced impact in Orthopaedic Surgery patients. Despite the reduction, both groups reported substantial and comparable percentages of unused opioids, indicating a need for targeted adjustments to minimize unused opioids.

## Level of evidence

IV.

## Introduction

The ongoing opioid epidemic in the United States presents a significant public health challenge. Surgery and the perioperative period, with its inherent pain management needs, remain a critical window for potential opioid dependence among previously naive patients [1,2]. Recognizing this vulnerability, initiatives such as Enhanced Recovery After Surgery (ERAS) protocols have emerged as promising strategies to minimize opioid use while providing effective pain control [3–7].

This study investigates the effectiveness of the "Billion Pill Pledge", a comprehensive ERAS program launched across a number of hospitals across Iowa, with the goal of reducing postoperative opioid prescription rates. Existing literature largely evaluates the use of enhanced recovery pathways in cohorts of patients undergoing the same surgical procedure [1,2,4,6–10]. This study instead assesses ERAS pathways designed for implementation across various surgical specialties. Utilizing patient-reported data alongside pre-ERAS implementation opioid quantities, the study examined the program's impact on opioid prescribing in the postoperative window.

The study hypothesized that after implementation of the ERAS protocol, fewer opioids would be prescribed postoperatively. The study also aimed to identify any differences in prescribing habits, both before and after "Billion Pill Pledge" implementation, between Orthopaedic Surgery and General Surgery. Secondarily, proportions of prescribed opioids that remained unused were also of interest, and were compared between surgical specialties. The potential for excessive postoperative prescribing to contribute to communal drug diversion is well studied [2,11,12]. Reducing this risk is a focal metric in measuring the broader success of opioid-minimizing pain management strategies

## Methods

### Patient selection

Institutional review board (IRB) approval was received for retrospective chart review of patients across 9 Iowa hospitals participating in the "Billion Pill Pledge" program. Patient data was accessed on September 1st 2024 through a deidentified patient database provided by Goldfinch Health Incorporated. Patients who underwent surgery between the dates November 2022 and November 2023 were identified. From the participating sites, patients from 18 to 80 years old who underwent Orthopaedic Surgery or General Surgery procedures were included in this study. Across the entire cohort, procedures were carried out by a total of 24 different surgeons. Inclusion criteria were all surgical patients participating in the ERAS protocols who had provided data on pill usage post discharge. Exclusion criteria were surgical specialties other than Orthopaedic Surgery or General Surgery, prior use of data in a study, or incomplete data for pill counts after discharge.

Due to a relative abundance of Orthopaedic Surgery data, when determining cohort size, an allocation ratio of N2/N1 = 0.5 was used (where, N2/N1 = General Surgery cohort size/ Orthopaedic Surgery cohort size). Using mean (range) prescription counts across a preliminary sample of the larger deidentified patient database provided by Goldfinch Health Inc, an a priori analysis identified a cohort size with sufficient power of 80% to identify an effect size of at least 0.35 at $\alpha = 0.05$.

### Measuring opioid quantities

For each patient, a pre-ERAS dataset and a post-ERAS dataset were extracted from the deidentified database retrospectively screened in this study. Post-ERAS data was prospectively reported by patients and noted into the database by hospital staff, as per updates to hospital standing orders. Meanwhile, pre-ERAS data input into the database were surgeon generated estimates of the postoperative opioid quantity the patient would have received prior to implementation of the Billion Pill Pledge. These hypothetical estimates of pre-ERAS prescription sizes were made by the surgeons treating each respective patient. The estimates were made based on the surgeons' prescribing patterns for similar procedure types prior to the launch of the Billion Pill Pledge.

Post-ERAS datasets also included opioid consumption data, which had been collected by nursing staff by way of patient follow up over the phone. Variables measured included prescribed quantity, quantity utilized, and remaining quantity. Pre-ERAS quantities were compared with newly captured post-ERAS quantities to gauge the efficacy of the program in minimizing postoperative opioid prescriptions.

All opioid counts were initially measured and reported in absolute pills. Opioid type and dosage in milligrams for each respective patient were then used to convert data to MMEs in accordance with established conversion factors [13]

### Statistical analyses

The Shapiro-Wilk test was used to evaluate continuous data for normality. Nonnormally distributed data were then reported as means (range). Statistical difference between pre-ERAS versus post-ERAS means of the same group was evaluated using the Wilcoxon-Signed Rank test. Means for different groups (i.e., Orthopaedic versus General Surgery) at the same time point were compared using the Mann-Whitney U test. Values of $p < .05$ were considered statistically significant and all tests comparing means were two tailed. When considering noncontinuous data, chi-squared tests were used to analyze associations in bivariate comparisons. All analyses were performed using IBM SPSS Version 29 (IBM Corp., Armonk, N.Y.).

The study cohort was split into groups based on specialty of surgical care, namely Orthopaedic Surgery and General Surgery cohorts. Variables for each cohort were pre-ERAS prescribed quantity, post-ERAS prescribed quantity, quantity utilized, and remaining quantity. For each patient, a percent reduction in prescription was calculated by taking

the difference between pre-ERAS and post-ERAS prescription size as a percentage of the pre-ERAS prescription size. Furthermore, for each patient, percent of MME's leftover was calculated by taking the remaining quantity of MMEs as a percentage of the post-ERAS prescribed quantity. Means (range) were calculated for each of these variables and percentages. Statistical comparisons of means were carried out using the tests listed above (i.e., Wilcoxon-Signed Rank and Mann-Whitney U tests). Noncontinuous data measured included refill rates, the incidence of zero opioids consumed by a patient, and the incidence of zero opioids remaining reported by a patient. These were each compared between specialties using chi-squared tests or fisher's exact tests as needed.

### Implementation of billion pill pledge/ERAS pathways

Each participating hospital was partnered with Goldfinch Health Inc (Austin, TX). Facilities and standing orders at each partner site were updated to agree with the "Billion Pill Pledge" ERAS pathway as detailed below.

1. *Before Surgery*: Patients received comprehensive education on pain associated with surgery in order to manage expectations for recovery. Preoperative hydration was provided 2 hours before surgery using ClearFast or Gatorade. To prevent inflammation, Meloxicam 10 mg or Celecoxib 400 mg was administered, while nerve pain prophylaxis was achieved through Pregabalin 75 mg (preferred) or Gabapentin 300 mg (alternative). Preemptive analgesia was achieved with Tylenol use (1000 mg daily for two days before surgery).

2. *Perioperative/Intraoperative*: Surgery was made to prioritize minimally invasive techniques, as well as an outpatient surgical setting to reduce hospital stays. Neuraxial anesthesia and sedation were used to ensure intraoperative comfort, while long-acting local anesthesia in the surgical field (i.e., Liposomal Bupivicaine) provided targeted pain relief at the site of incision. Decadron, Zofran, and/or Scopolamine patches were routinely administered to prevent postoperative nausea. In procedures requiring general anesthesia, Sugammadex was used as a reversal agent.

3. *After Surgery*: First line pain management focused on multimodal strategies based around non-opioid medications. The multimodal agents included Tylenol, selective Cox-2 NSAIDs (i.e., Meloxicam or Celecoxib), Pregabalin or Gabapentin, Robaxin (when called for, based on surgical procedure), and muscle relaxants such as Flexeril. Opioid prescriptions were limited to a maximum of 10 doses, which were restricted for severe breakthrough pain. Supplemental steps included routine ice application and early oral intake, including chewing gum, to promote gastrointestinal motility.

Before the date of surgery, "Prepared for Surgery Tool Kits" were delivered to each patient's home. Kits included hot/cold packs, a complex carbohydrate pre-surgical drink, post-surgical chewing gum, a drug disposal kit, and patient education materials. Postoperative nursing follow-ups provided a second opportunity for patient education (Fig 1).

## Results

### Cohort overview

This study included a total of 180 eligible patients for whom procedure classification and complete pill usage data were available. Demographic information of the patient cohort was not evaluated, nor was it available as the data analyzed in this study was retrieved from a deidentified patient database provided by Goldfinch Health. Patients in the cohort had undergone surgeries including: Knee arthroplasty (37%), Other Orthopaedic Surgery (21%), Hernia (21%), Cholecystectomy (9%), Hip Arthroplasty (9%), and Other General Surgery (3%) (Fig 2).

### Prescription sizes pre-ERAS versus post-ERAS

Across 180 patients for whom complete data was collected, mean (range) opioids prescribed postoperatively decreased to 151 MMEs (range: 25–2400 MMEs) post-ERAS implementations from a prior mean of 341 MMEs (range: 25–7200 MMEs) pre-ERAS implementation (p < .001).

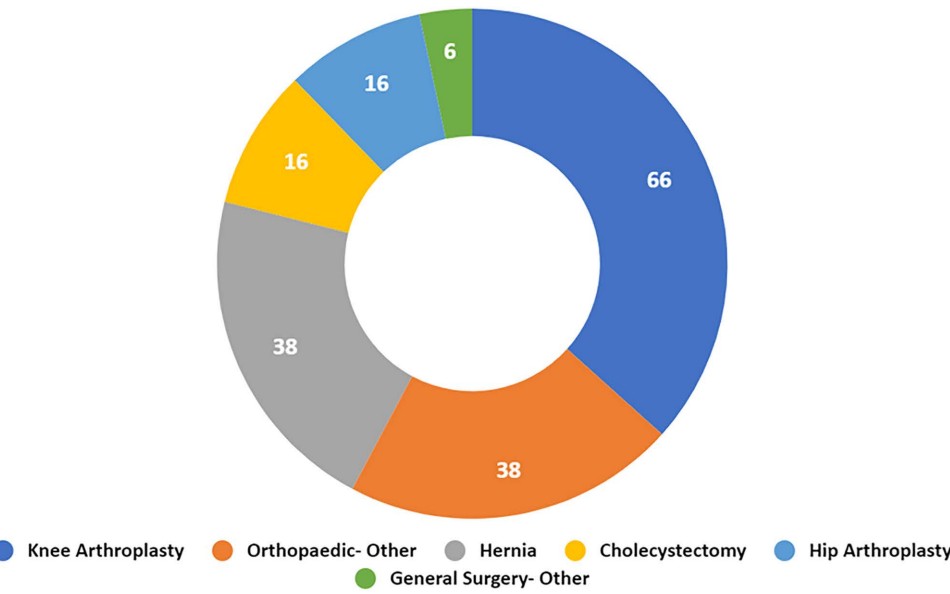

**Fig 1. Guidelines set by the Billion Pill Pledge protocol in practice.**

**Knee Arthroplasty** ● **Orthopaedic- Other** ● **Hernia** ● **Cholecystectomy** ● **Hip Arthroplasty**
● **General Surgery- Other**

**Fig 2. Breakdown of Cohort by Procedure Type: knee arthroplasty (n = 66), orthopaedic- other (n = 38), hernia (n = 38), cholecystectomy (n = 16), hip arthroplasty (n = 16), general surgery- other (n = 6).**

Orthopaedic Surgery patients saw a mean 45% decrease in opioids prescribed from a mean 462 MMEs (range: 50–7200 MMEs) pre-ERAS to mean 197 MMEs (range: 25–2400 MMEs) post-ERAS implementation (p < .001). General Surgery patients saw a mean 38% decrease in prescription size from 100 MMEs (range: 25–150 MMEs) to 60 MMEs (range: 25–150 MMEs) post-ERAS implementation (p < 0.001). Across the total cohort, a mean 43% decrease in MMEs prescribed was seen post-ERAS, relative to the pre-ERAS baseline (Table 1).

Pre-ERAS implementation, mean prescription size for the Orthopaedic Surgery cohort was found to be significantly greater than that for the General Surgery cohort (462 vs 100 MMEs) (p < .001) (Fig 3). Post-ERAS implementation, this

**Table 1. Prescription data.**

| | Orthopaedic Surgery | General Surgery | p-value |
|---|---|---|---|
| No. of patients | 120 | 60 | |
| Mean MMEs prescribed pre-ERAS | 462 | 100 | <.001 |
| Mean MMEs prescribed post-ERAS | 197 | 60 | <.001 |
| Mean % reduction from pre-ERAS | 45 | 38 | .002 |

Specialty of Surgical Care With Corresponding Mean Pre-ERAS and Post-ERAS Opioid Prescription Quantity, and Mean Percent Reduction in Prescriptions.

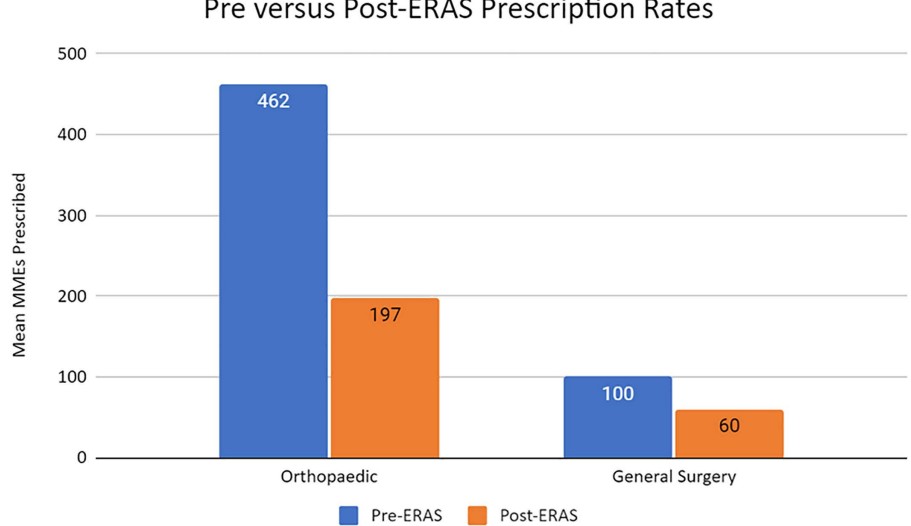

**Fig 3. Bar graph comparing mean Pre-ERAS and Post-ERAS opioid prescription rates by Surgical specialty.**

trend continued, as Orthopaedic Surgery mean prescription rates remained higher than General Surgery prescription rates (197 vs 60 MMEs) (p < .001) (Fig 3). Upon statistical analysis, the Orthopaedic Surgery group was found to have a larger mean percent reduction in prescriptions relative to the General Surgery group (45% versus 38%) (p = .002) (Table 1).

## Leftover MMEs by specialty

Out of 27,218 MMEs prescribed across 180 patients, 12,090 (44%) went unused. The Orthopaedic Surgery cohort displayed a mean (range) quantity utilized of 126 MMEs (range: 0–2400 MMEs), versus 26 MMEs (range: 0–150 MMEs) in the General Surgery cohort (p < .001). There was no statistical difference found between the two groups in terms of mean unused opioids remaining, with a mean 83 MMEs (range: 0–1860 MMEs) remaining and mean 36 MMEs (range: 0–150 MMEs) remaining in the Orthopaedic Surgery and General Surgery cohorts, respectively (p = .06). Similarly, no statistical difference was found in the mean percent of prescribed MMEs leftover by each group (p = .07) (Table 2).

## Consumption and refill trends after ERAS implementation

Across the entire cohort, 42 patients consumed no opioids after ERAS implementation. The incidence rate of zero MMEs consumed was 21/120 (18%) in the Orthopaedic Surgery cohort versus 21/60 (35%) in the General Surgery cohort. A

**Table 2. Opioid consumption data.**

|  | Orthopaedic Surgery | General Surgery | p-value |
|---|---|---|---|
| No. of patients | 120 | 60 |  |
| Mean quantity utilized | 126 | 26 | <.001 |
| Mean remaining quantity | 83 | 36 | .06 |
| Mean % MMEs leftover | 47 | 59 | .07 |

Patients sorted by specialty of care with corresponding mean opioids consumed, mean opioids unused, and mean % of prescribed MME's leftover.

chi-squared test of independence found a significant relation between surgical specialty and incidence of an opioid consumption free postoperative period χ2 (1, *N* = 180) = 6.8478, p = .009 (Table 3).

Similarly, a chi-squared test of independence detected a significant relation between surgical specialty of care and incidence of a postoperative period with zero opioids left unused, χ2 (1, *N* = 180) = 10.1671, p = .00143. Zero unused opioids remaining were reported by 38/120 (32%) of patients in the Orthopaedic Surgery cohort, and 6/60 (10%) of patients in the General Surgery cohort (Table 3).

Only 9 patients requested refills to their postoperative opioid prescription resulting in a 5% refill rate across the total cohort of 180 patients. Incidence rate for opioid refill was 8 out of 120 (7%) and 1 out of 60 (2%) for the Orthopaedic Surgery and General Surgery cohorts, respectively (Fig 4). The results of a Fisher exact test (p = 0.3) did not indicate a significant association between incidence of a prescription refill and surgical specialty of care.

## Discussion

"The Billion Pill Pledge" ERAS pathways are currently implemented across 9 hospitals in Iowa, irrespective of surgical classification. This study aimed to evaluate the efficacy of these pathways across two distinct surgical specialties. The study's findings upheld the study hypothesis with a significant decrease in mean postoperative prescriptions across the entire cohort after the program's implementation, with a significantly greater drop seen in the Orthopaedic Surgery cohort than that seen in the General Surgery cohort. However, both groups also contributed to high amounts of unused opioids.

Based on prior hospital data, both groups presented different mean pre-ERAS prescription sizes. The Orthopaedic Surgery cohort had a higher mean pre-ERAS prescription by a drastic margin at 462 MMEs (range: 50–7200 MMEs). This was significantly higher than the mean pre-ERAS prescription for the General Surgery cohort at 100 MMEs (range: 25–150 MMEs). These results mirror past studies that have established an existing trend of higher opioid prescribing in Orthopaedic Surgery [14–16]. These findings also highlight the need for more targeted focus on Orthopaedic Surgeons and their patients to avoid opioid abuse and diversion to their broader community.

**Table 3. Noncontinuous data on opioid use.**

|  | Orthopaedic Surgery | General Surgery | p-value |
|---|---|---|---|
| No. of patients | 120 | 60 |  |
| Zero opioids consumed | 21 | 21 | .009 |
| Zero opioids remaining | 38 | 6 | .001 |
| Refills | 8 | 1 | .3 |

Noncontinuous data examining relation between specialty of care and likelihood of zero opioids consumed, zero opioids left unused, and patient refill request. All data pertains particularly to postoperative opioid consumption.

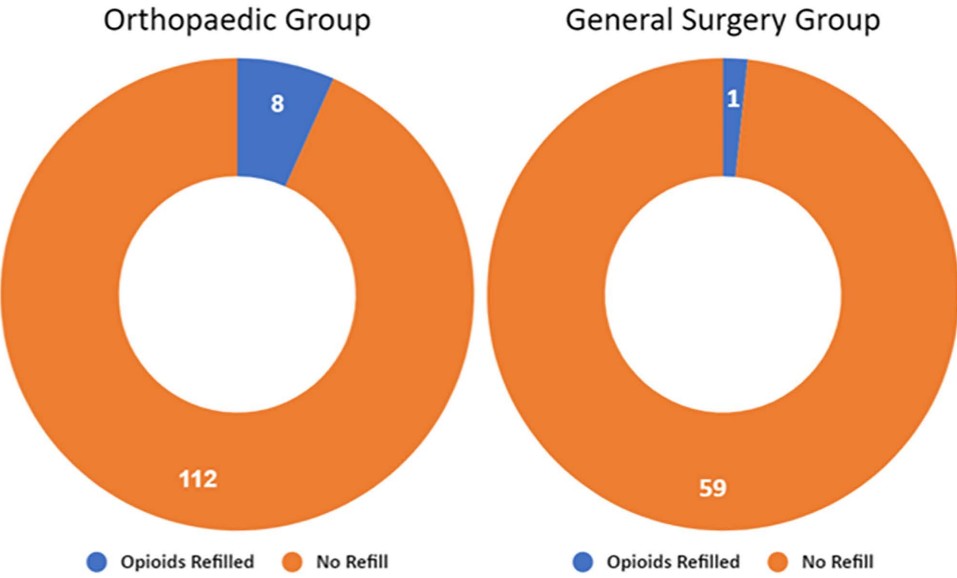

**Fig 4. Refill rate by surgical specialty.**

The "Billion Pill Pledge" program's ERAS pathway successfully reduced postoperative opioid prescribing across both surgical specialties. Mean percent reduction of prescription size observed in the Orthopaedic Surgery cohort was significantly greater than that observed in the General Surgery cohort. Differences in impact of these ERAS pathways on the two cohorts could perhaps be attributed to the higher homogeneity of patient demographics, procedures, and outcomes seen in elective surgery. These findings warrant further exploration into which specialties may inherently be more suited to ERAS pathways. Moreover, that prescription sizes in both cohorts were significantly reduced could be a result of aspects of the ERAS pathways that may affect patient outcomes in a broader sense, independent of the kind of procedure being carried out, for instance longitudinal patient education. Studies have and should continue to explore whether particular aspects of these pathways are responsible for larger relative portions of their overall impact [10]. In addition, exploring the applicability of the ERAS pathways to other surgical specialties is also paramount. For example, reducing opioid dependence in Gynecologic Surgery remains a crucial public health issue given previous findings that a high proportion of Gynecologic Surgery patients demonstrate high persistent opioid use post-operatively [17].

A total of 136 patients in this study reported leftover unused opioids. This constitutes 76% of the study's total cohort of 180 patients. These results are consistent with previously observed rates of 67% to 92% of surgical patients reporting unused opioids postoperatively [11]. Of the opioids prescribed in the total cohort, 44% went unused. This is in concordance with previous counts of opioids unused postoperatively varying from as low as 42% to as high as 73% [11,12]. Notably, this study's results found no significant difference in mean percent of MMEs leftover by General Surgery versus Orthopaedic Surgery patients (59% vs 47%). It is well-documented that most surgical patients do not dispose of leftover medication [18]. In accordance with the "Billion Pill Pledge" ERAS pathway, patients were provided with drug disposal bottles, as well as educated on their use by the nursing staff before and after surgery. Though patient use, or lack thereof, of provided disposal kits was not documented, studies have shown that even providing patients with charcoal waste bags can increase the odds of unused medication disposal manifold [19]. In future studies, gauging the effectiveness of drug disposal systems will be an important point of focus in determining the risk of postsurgical opioid diversion in patient communities.

Future ERAS protocols should continue to be designed with particular attention given to pill usage at discharge, especially given the potential of this time point as a window for drug diversion and persistent use by patients. Existing literature largely focuses on pain management protocols designed with a surgical field or even a particular surgery in mind [6–9,20]. In contrast, the "Billion Pill Pledge" ERAS protocols considered in this study have been implemented with a breadth of surgical procedures in mind and across multiple hospitals in Iowa. Prior studies have shown that the effectiveness of ERAS protocols varies widely by the surgical procedure, with certain protocols yielding drastic reduction in opioid use, and others resulting in no change in opioid use post-ERAS implementation [6–9,20]. Seeing how in this study Orthopaedic Surgery patients displayed a greater mean percent reduction in prescription size than their General Surgery counterparts did, the possibility of tailoring these ERAS protocols to individual procedures warrants further study. Given that both specialties saw a significant reduction in prescription sizes however, equally deserving of investigation is the expansion of these same protocols to additional surgical specialties that may benefit the same amount as the two considered in this study.

This study had several limitations. First, the Pre-ERAS prescription quantity for each patient was provided as an estimate by surgeons based on their previous record of prescribing before ERAS implementation. Also, unlike similar studies, this study's cohort was not split into a control and experimental group. Additionally, there was a lack of Pre-ERAS data on unused opioids. Since there was no Pre-ERAS benchmark for comparison in this regard, the study was unable to assess the utility of the "Billion Pill Pledge" ERAS pathway's implementation in reducing MMEs left over.

Additional limitations exist around patient information. Data analyzed in this study was retrieved from a deidentified patient database provided by Goldfinch Health (Austin, TX). Therefore, analysis did not include nor account for the demographics of the patient population. This study did not account for prior history of opioid use in patients- a potential confounder with possible effects on opioid consumption habits in patients. The impact of postoperative pain management interventions in chronic opioid users versus opioid naïve patients is a topic warranting further investigation. Due to a lack of the amount of data required to produce a statistically powered comparison between procedure types, this study did not account for the different procedures included in each surgical specialty. Data collected in the future should include not only a broad range of surgical specialties, but also a breadth of surgeries within each field. Comparisons of ERAS efficacy in different surgical procedures belonging to the same surgical specialty is another point of interest warranting the attention of future research.

## Conclusion

In 9 hospitals across Iowa, the "Billion Pill Pledge" ERAS pathway significantly reduced postoperative opioid prescribing across General Surgery and Orthopaedic Surgery patients, with a more pronounced effect in Orthopaedic Surgery. That the program successfully reduced opioid prescribing in both specialties suggests that the protocols implemented are suited for use across a greater breadth of surgical specialties and procedures. Future research should address the disposal of unused opioids, expand to other surgical specialties, and identify particular procedures that may be in critical need of opioid prescription control. Overall, the "Billion Pill Pledge" ERAS program demonstrates promise for reducing postoperative opioid use, highlighting the need for continued refinement and expansion into other regions and surgical specialties.

## Author contributions

**Conceptualization:** Wali U. Pirzada, Simran Shamith, Roland N. Leyson, Sara A. Khan, Sina Ramtin, Asif M. Ilyas.

**Data curation:** Wali U. Pirzada, Simran Shamith, Roland N. Leyson, Sara A. Khan, Asif M. Ilyas.

**Formal analysis:** Wali U. Pirzada, Simran Shamith, Roland N. Leyson, Sara A. Khan, Asif M. Ilyas.

**Investigation:** Simran Shamith, Roland N. Leyson, Sara A. Khan, Sina Ramtin, Asif M. Ilyas.

**Methodology:** Wali U. Pirzada, Simran Shamith, Roland N. Leyson, Sara A. Khan, Sina Ramtin, Asif M. Ilyas.

**Software:** Wali U. Pirzada, Simran Shamith, Roland N. Leyson, Sara A. Khan, Sina Ramtin, Asif M. Ilyas.

**Supervision:** Sina Ramtin, Asif M. Ilyas.

**Validation:** Wali U. Pirzada, Sina Ramtin.

**Writing – original draft:** Wali U. Pirzada.

**Writing – review & editing:** Wali U. Pirzada, Simran Shamith, Roland N. Leyson, Sara A. Khan, Sina Ramtin, Asif M. Ilyas.

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
