## [Decision Letter · Decision Letter 0]

4 Jul 2025

Dear Dr. Ramtin,

We look forward to receiving your revised manuscript.

Kind regards,

Moises Auron, MD, FAAP, FACP, SFHM, FRCP (Lon), FRCPCH

Academic Editor

PLOS ONE

Journal Requirements:

2. In the online submission form, you indicated that:

“The data underlying the results presented in the study are available from Goldfinch Health Incorporated, who can be contacted at https://www.goldfinchhealth.com/”

3. Uploaded as supplementary information.

**Additional Editor Comments:**

Please kindly appraise the comments by the reviewer and resubmit with the appropriate changes.

Reviewers' comments:

Reviewer's Responses to Questions

**Comments to the Author**

1. Is the manuscript technically sound, and do the data support the conclusions?

Reviewer #1: Yes

2. Has the statistical analysis been performed appropriately and rigorously?

Reviewer #1: I Don't Know

3. Have the authors made all data underlying the findings in their manuscript fully available?

Reviewer #1: No

4. Is the manuscript presented in an intelligible fashion and written in standard English?

Reviewer #1: Yes

Reviewer #1: Great work at demonstrating that common sense pain practices help in reducing opioid prescriptions and usage.

A few minor questions remain that would make this manuscript more informative.

If you could define the pre-ERAS implementation opioid quantity more granularly that would be good. In your discussion you reported that the surgeons gave you an estimate of how much they would prescribe before. How did they determine that? Did each of the surgeons or their practices have an opioid prescription protocol for individual procedures that they would follow? If they did that would be a much more reliable number than an estimate and you could put that into the manuscript.

How did you retrospectively get patient reported measures regarding pain medication usage on a post op visit? When you said you validated that with the providers is it all from chart review? If the Billion Pill Pledge enrollment required documentation enhancement to be able to reliably capture that data it would be wise to mention that in the manuscript.

In figure 1- are all ERAS protocols the same for any and all surgeries barring what your mentioned? Is celecoxib/meloxicam a one time use before surgery? In other words a patient getting a knee replacement will get the same 10 dose opioid script as someone who had a diagnostic arthroscopy? Also either use Tylenol or Acetaminophen (just a minor detail)

Could you get a breakdown of inpatient versus outpatient surgeries on your dataset? If inpatient surgeries how did you capture post op opioid use while the patient was hospitalized. Did you have access to their inpatient medication administration record? What was the average length of stay of inpatient surgeries?

You have a breakdown of the kind of surgeries that were performed, can we get MMEs associated with each of the individual surgeries broken down in a table instead of Orthopedics vs General Surgery? Similarly if you have data for left over opioids for each of these procedures, you may be able to predict which type of surgical procedure patients need a full 10 dose script versus who doesn't?

Finally, does the dataset capture prior history of opioid use or chronic opioid use in the patients?

**Do you want your identity to be public for this peer review?** For information about this choice, including consent withdrawal, please see our Privacy Policy

Reviewer #1: No

---

## [Author Response · Author response to Decision Letter 1]

23 Aug 2025

Reviewer# 1 Comments:

1. Great work at demonstrating that common sense pain practices help in reducing opioid prescriptions and usage.

If you could define the pre-ERAS implementation opioid quantity more granularly that would be good. In your discussion you reported that the surgeons gave you an estimate of how much they would prescribe before. How did they determine that? Did each of the surgeons or their practices have an opioid prescription protocol for individual procedures that they would follow? If they did that would be a much more reliable number than an estimate and you could put that into the manuscript.

Response: We sincerely thank the reviewer for their positive feedback to our manuscript as well as for bringing their concern to our attention. Pre-ERAS prescription sizes used in this study were surgeon-generated estimates of the amount of opioids the patient would have theoretically been prescribed postoperatively had they been treated prior to the launch of the Billion Pill Pledge. Each patient’s pre-ERAS estimate was generated by the respective surgeon in charge of their care. Surgeons based these estimates off their prior prescribing habits for similar procedure types as indicated by hospital records. These pre-ERAS hypotheticals were input into a deidentified database of surgical patients cared for at BPP partner sites. It was this database that was retrospectively screened for the purpose of this study. The portion titled “Measuring opioid quantities” under our methods section has been revised to better illustrate this workflow.

2. How did you retrospectively get patient reported measures regarding pain medication usage on a post op visit? When you said you validated that with the providers is it all from chart review? If the Billion Pill Pledge enrollment required documentation enhancement to be able to reliably capture that data it would be wise to mention that in the manuscript.

Response: We thank the reviewer for bringing this to our attention. Upon initiation of the Billion Pill Pledge, standing orders at all partner sites were updated to instruct nursing staff to prospectively collect opioid prescription and consumption data via patient follow up over the phone. This data was documented into a deidentified database, which was retrospectively screened for the purpose of this study. The portion titled “Measuring opioid quantities” under our methods section has been revised to better illustrate this workflow.

3. In figure 1- are all ERAS protocols the same for any and all surgeries barring what your mentioned? Is celecoxib/meloxicam a one time use before surgery? In other words a patient getting a knee replacement will get the same 10 dose opioid script as someone who had a diagnostic arthroscopy? Also either use Tylenol or Acetaminophen (just a minor detail)

Response: We thank the reviewer for highlighting their concerns. Upon initiation of the Billion Pill Pledge, standing orders at each partner site were updated to implement this pathway for all surgeries performed. Exceptions were made only as necessary, at the surgeon’s discretion. While celecoxib/meloxicam use is noted as a “Before Surgery” intervention, it may also be used as the selective Cox-2 NSAID of choice for pain management “After Surgery”, as per physician’s discretion. This has been listed in the “Implementation of billion pill pledge/ERAS pathways” portion of the methods section. We have also made revisions to clarify that each patient was limited to a maximum of 10 opioid doses. While different patients may have been prescribed different quantities of opioids, the maximum quantity prescribable as per updated standing orders was 10 doses. This opioid script was meant to be reserved for severe breakthrough pain, while the first line of pain management postoperatively focused on non-opioid medications. We have also revised the manuscript to replace “Acetaminophen” with “Tylenol”.

4. Could you get a breakdown of inpatient versus outpatient surgeries on your dataset? If inpatient surgeries how did you capture post op opioid use while the patient was hospitalized. Did you have access to their inpatient medication administration record? What was the average length of stay of inpatient surgeries?

Response: We thank the reviewer for sharing their concerns. In compliance with the Billion Pill Pledge, all surgeries were performed in the outpatient setting. The deidentified database screened in this study therefore does not contain any inpatient medication records. The purpose of this study was to track only postoperative opioid medications made at discharge (or after). We have revised the methods section to more explicitly state this.

5. You have a breakdown of the kind of surgeries that were performed, can we get MMEs associated with each of the individual surgeries broken down in a table instead of Orthopedics vs General Surgery? Similarly if you have data for left over opioids for each of these procedures, you may be able to predict which type of surgical procedure patients need a full 10 dose script versus who doesn't?

Response: We thank the reviewer for bringing this concern to our attention. The deidentified database screened in this study did indeed include information on procedure type. However, we were unable to acquire cohort sizes large enough to perform a statistically powered comparison between all procedure types. Recognizing the value of a statistically powered analysis, we decided to base the study instead on a Cross-specialty comparison. We have acknowledged this constraint in the limitations paragraph of our discussion section. We have also revised the methods section to more explicitly present the findings of our a priori analysis.

6. Finally, does the dataset capture prior history of opioid use or chronic opioid use in the patients?

Response: We thank the reviewer for highlighting their concern. Unfortunately, the database screened in this study contains no patient information other than procedure type, opioid prescriptions, and opioid consumption data. This prevents the study design from accounting for opioid consumption habits in opioid naïve patients versus chronic opioid users. We have included this constraint as one of the limitations of this study in the discussion section.

---

## [Editor Report · Decision Letter 1]

28 Aug 2025

Utility of enhanced recovery after surgery protocols in reducing postoperative opioid use across different surgical specialties – an analysis of Iowa’s billion pill pledge program

PONE-D-25-08723R1

Dear Dr. Ramtin,

We’re pleased to inform you that your manuscript has been judged scientifically suitable for publication and will be formally accepted for publication once it meets all outstanding technical requirements.

Kind regards,

Moises Auron, MD, FAAP, FACP, SFHM, FRCP (Lon), FRCPCH

Academic Editor

PLOS ONE

Additional Editor Comments (optional):

Thank you for your submission and eloquent response to the reviewers queries.
---

## [Editor Report · Acceptance letter]

PONE-D-25-08723R1

PLOS ONE

Dear Dr. Ramtin,

I'm pleased to inform you that your manuscript has been deemed suitable for publication in PLOS ONE. Congratulations! Your manuscript is now being handed over to our production team.

Kind regards,

on behalf of

Dr. Moises Auron

Academic Editor

PLOS ONE